# Virus–host interactions in carcinogenesis of Epstein-Barr virus-associated gastric carcinoma: Potential roles of lost ARID1A expression in its early stage

Hiroyuki Abe[1], Akiko Kunita[1], Yuya Otake[1,2], Teru Kanda[3], Atsushi Kaneda[4], Tetsuo Ushiku[1], Masashi Fukayama[1,5]*

1 Department of Pathology, Graduate School of Medicine, The University of Tokyo, Tokyo, Japan,
2 Department of Pathology, Nippon Life Hospital, Osaka, Japan, 3 Department of Microbiology, Tohoku Medical and Pharmaceutical University, Sendai, Japan, 4 Department of Molecular Oncology, Graduate School of Medicine, Chiba University, Chiba, Japan, 5 Asahi TelePathology Center, Asahi General Hospital, Chiba, Japan

* mfukayama-tky@umin.org

**Data Availability Statement:** All relevant data are within the manuscript.

## Abstract

Epstein–Barr virus (EBV)-associated gastric carcinoma (EBVaGC) is a distinct molecular subtype of gastric cancer characterized by viral infection and cellular abnormalities, including loss of AT-rich interaction domain 1A (ARID1A) expression (lost ARID1A). To evaluate the significance of lost ARID1A in the development of EBVaGC, we performed *in situ* hybridization of EBV-encoded RNA (EBER) and immunohistochemistry of ARID1A in the non-neoplastic gastric mucosa and intramucosal cancer tissue of EBVaGC with *in vitro* infection analysis of *ARID1A*-knockdown and -knockout gastric cells. Screening of EBER by *in situ* hybridization revealed a frequency of approximately 0.2% EBER-positive epithelial cells in non-neoplastic gastric mucosa tissue samples. Six small foci of EBV-infected epithelial cells showed two types of histology: degenerated (n = 3) and metaplastic (n = 3) epithelial cells. ARID1A was lost in the former type. In intramucosal EBVaGC, there were ARID1A-lost (n = 5) and -preserved tumors (n = 7), suggesting that ARID1A-lost carcinomas are derived from ARID1A-lost precursor cells in the non-neoplastic mucosa. Lost ARID1A was also observed in non-neoplastic mucosa adjacent to an ARID1A-lost EBVaGC. *In vitro* experiments using siRNA knockdown and the CRISPR/Cas9-knockout system demonstrated that transient reduction or permanent loss of *ARID1A* expression markedly increased the efficiency of EBV infection to stomach epithelial cells. Taken together, lost ARID1A plays a role in initiating EBV-driven carcinogenesis in stomach epithelial cells, which develop to a distinct subtype of EBVaGC within the proper mucosal layer. Lost ARID1A is one of the constituents of virus–host interactions in the carcinogenesis of EBVaGC.

**Funding:** This study was supported by Grants-in-Aid for Scientific Research (KAKENHI) from the Japan Society for the Promotion of Science (https://www.jsps.go.jp/english/) [grant numbers 26253021 (M.F.) and 18K07012 (H.A.)], and by the AMED-CREST program from the Japan Agency for Medical Research and Development (https://www.amed.go.jp/en/index.html) [grant number 16gm0510010h0005 (A.K. and M.F.)]. The funders had no role in study design, data collection and analysis, decision to publish, or preparation of the manuscript.

**Competing interests:** The authors have declared that no competing interests exist.

**Abbreviations:** ARID1A, AT-rich interactive domain 1A; EBERs, EBV-encoded RNAs; EBV, Epstein-Barr virus; EBVaGC, EBV-associated gastric cancer; MSI, microsatellite instability.

## Introduction

Epstein–Barr virus (EBV)-associated gastric carcinoma (EBVaGC) is one of four molecular subtypes of gastric cancer. The other subtypes include microsatellite instable (MSI), chromosomal unstable, and genetically stable gastric cancer [1]. EBVaGC is defined as the clonal expansion of EBV-infected cells and is characterized by epigenetic and genetic abnormalities, such as the extensive DNA methylation of promoter regions and recurrent mutations of several genes. One example is the mutation of AT-rich interaction domain 1A (*ARID1A*) [1, 2], which encodes a subunit protein of the SWI/SNF chromatin remodeling complex. Loss of ARID1A expression (lost ARID1A) has been observed in cancers of the ovary, endometrium, and colon [3–7], and mutation of the *ARID1A* gene is closely associated with the loss of protein expression. In gastric cancer, *ARID1A* is frequently mutated in two molecular subtypes, EBVaGC and MSI [1, 2]. While insertion/deletion mutations caused by frequent slippage on microsatellite sequences are predominant in the MSI subtype [8–10], nonsense mutations are distributed throughout the coding region in EBVaGC [1, 2]. In our previous study using ARID1A immunohistochemistry, the frequency of lost ARID1A was similarly high in early and advanced EBVaGC tumors but was increased in the advanced tumors of other gastric cancer subtypes [11]. We also revealed that EBV infection did not induce the loss of ARID1A expression [11]. Based on these results, we hypothesized that ARID1A expression is lost at an early stage of carcinogenesis in EBVaGC, or in other words, within non-neoplastic glands [12, 13].

To clarify the interactions between EBV infection and host abnormalities in the stomach mucosa at different phases of carcinogenesis of EBVaGC, the following questions were investigated: 1) whether lost ARID1A coexists with EBV infection in non-neoplastic glands; 2) whether lost ARID1A affects the pathology of EBVaGC within the proper mucosal layer; and 3) whether lost ARID1A promotes the EBV infection of stomach epithelial cells. To address these questions, we first screened EBV-infected epithelial cells in a large number of non-neoplastic stomach mucosa tissue sections through *in situ* hybridization (ISH) of EBV-encoded RNAs (EBERs). These small foci were evaluated by ARID1A immunohistochemistry to identify the co-existence of EBER-positivity and lost ARID1A. Next, we compared the morphology of intramucosal EBVaGC between ARID1A-lost or -preserved tumors to evaluate the effect of lost ARID1A at the stage of intramucosal carcinoma. Using *ARID1A*-knockdown and -knockout systems, we demonstrated that lost ARID1A increased the EBV infection of stomach epithelial cells *in vitro*. This study aimed to elucidate the complex interactions between the virus and host in EBVaGC carcinogenesis, in addition to "cross-species DNA hypermethylation" [14] and "cross-species chromosomal interactions" [15] in this specific subtype of gastric cancer infected with EBV.

## Materials and methods

### Tissue samples

For the screening of EBER-positive glands in nonneoplastic gastric mucosa, we used three series of samples. The first one consisted of formalin-fixed and paraffin-embedded (FFPE) background mucosa tissues, which were surgically resected for gastric cancer treatment (1132 tissue blocks from 475 gastric cancer specimens) at The University of Tokyo Hospital from 2005 to 2007. The second series consisted of 1110 tissue blocks from 300 gastric cancer specimens (including 13 EBVaGC cases) resected at The University of Tokyo Hospital from 2010 to 2012. Of these, tissue blocks from the non-neoplastic mucosa of the lesser curvature were selected. In total, 2242 blocks were used for the screening of EBER-positive cells in this study.

The third source was derived from the file of one of the authors (TU), in which EBER-positive cells were identified in the non-neoplastic mucosa near the gastric cancer during the screening of EBV infection. More specifically, EBER-ISH was performed in approximately 300 cases of surgically or endoscopically resected gastric cancer from 1990 to 2020 which were suspicious for EBVaGC based on the morphological characteristics. In addition, 185 consecutive cases of endoscopically resected gastric cancer were screened by EBER-ISH.

EBVaGCs, in which invasion was limited within the proper mucosal layer (intramucosal carcinoma), were retrieved from the files of gastric carcinomas of the Department of Pathology, The University of Tokyo, that had been evaluated by EBER-ISH. Eleven cases were identified, all of which were surgically resected.

Tissue blocks of FFPE cancer tissue were used in this study. Each FFPE tissue block was sliced into 3-μm thick sections and used for EBER ISH and immunohistochemistry, as described below.

The study was approved by the institutional review board of the Graduate School of Medicine, The University of Tokyo (number G3521).

### *EBER* ISH and immunohistochemistry of ARID1A

EBER ISH was performed with a fluorescein isothiocyanate (FITC)-labeled EBER peptide nucleic acid (PNA) probe (Y5200; Dako, Glostrup, Denmark) and anti-FITC antibody (dilution 1:200; V0403; Dako). ARID1A immunohistochemistry was performed with an automated immunostainer Ventana BenchMark XT or ULTRA System (Roche, Basel, Switzerland) in accordance with the manufacturer's protocols using an anti-ARID1A antibody (dilution 1:100; rabbit polyclonal, HPA005456; Sigma-Aldrich, St. Louis, MO, USA) and OptiView DAB universal kit (Roche).

### Cell lines

We used two gastric cancer cell lines MKN74 and NUGC3, which are not infected with EBV. MKN74 (RCB1002) (RRID:CVCL_2791) was obtained from the Riken Bioresource Center Cell Bank (Tsukuba, Japan) and NUGC3 (JCRB0822) (RRID:CVCL_1612) was obtained from the Japanese Collection of Research Bioresources. The cell lines were cultured in RPMI-1640 supplemented with 10% fetal bovine serum, 40 units/mL penicillin, and 50 μg/mL streptomycin at 37˚C in a 5% $CO_2$ incubator. All human cell lines had been authenticated using short tandem repeat profiling within the previous three years. Cell lines were confirmed as mycoplasma-free by PCR.

### *ARID1A* knockdown

*ARID1A* knockdown was performed with a specific siGENOME SMARTpool (Thermo Fisher Scientific, Waltham, MA, USA), which is a pool of four different siRNA sequences. siRNA with no target gene (Thermo Fisher Scientific) was transfected as a control.

### *ARID1A* gene knockout

Optimal CRISPR/Cas9 targeting sites for *ARID1A* were identified using the Guide Design Resources of the Zhang Laboratory (http://crispr.mit.edu). crRNA, tracrRNA, and SpCas9 protein (Alt-R CRISPR-Cas9 System, Integrated DNA Technologies, CA, USA) were transfected into MKN74 cells as an RNP complex using RNAiMAX (Thermo Fisher). The crRNA sequence for *ARID1A* was as follows: 5'-AUGGUCAUCGGGUACCGCUGGUUUUAGAG-CUAUGCU-3'. After transfection and cloning, several clones were selected. The expression of

ARID1A was confirmed by western blotting, and two clones negative for ARID1A (#1 and #2) were used for further analyses.

## EBV infection of *ARID1A*-knockdown cells

The cell-to-cell contact method was used to infect *ARID1A*-knockdown cells with EBV. After 24 hours of incubation, recombinant EBV carrying the neomycin-resistant gene was infected using the cell-to-cell contact method [16]. The recombinant EBV-infected Burkitt lymphoma cell line Akata (gifted by Professor Kenzo Takada, Hokkaido University) was incubated with the gastric cancer cell lines for 48 hours, and then Akata cells were removed by washing with PBS. The gastric cancer cell lines were seeded into 15 cm-dishes and incubated with cell culture medium containing G418 (300 μg/mL for MKN74 and 200 μg/mL for NUGC3, Roche Diagnostics) for 10 to 14 days. As a negative control, the same gastric cancer cell lines were inoculated into 15-cm dishes and incubated with G418 for the same duration. All non-infected cells underwent cell death and subsequently disappeared. After incubation, cells were fixed with 4% paraformaldehyde and stained with hematoxylin to count the number of colonies.

## EBV infection of *ARID1A*-knockout cells

In this experiment, EGFP-EBV infection system [17] was applied to the two *ARID1A*-knockout MKN74 clones (#1 and #2) and the parental MKN74 cell line for evaluation of infection efficiency without G418 selection [18]. Briefly, AGS cells harboring EGFP-EBV were transfected with pSG5-BZLF1 using Lipofectamine 3000 (Thermo Fisher Scientific). Three days later, the culture supernatant was harvested and filtered through a membrane with 0.45-μm pores, and the filtrate was used as the virus solution. For infection, *ARID1A*-knockout MKN74 #1 and #2 clones or parental MKN74 cells were resuspended with 1 mL of the virus solution and incubated at 37˚C for 2 days. EBV-positive cells were observed under a fluorescence microscope (wavelength 525 nm; EVOS, Thermo Fisher Scientific). In addition, cells were analyzed by forward scatter, side scatter, and EGFP intensity with a flow cytometer (LSRFortessa X-20; BD Biosciences, CA, USA).

## RNA extraction and RT-PCR

Total RNA was extracted from gastric cancer cell lines with ISOGEN II (Nippon Gene, Toyama, Japan) according to the manufacturer's protocols. Extracted RNA was reverse transcribed with a ReverTra Ace qPCR RT Kit (Toyobo, Osaka, Japan), and real-time PCR was performed with a KAPA SYBR FAST qPCR Kit (Kapa Biosystems, Wilmington, MA, USA). The primers used were as follows: *ARID1A* forward, `CTTCAACCTCAGTCAGCTCCCA`; *ARID1A* reverse, `GGTCACCCACCTCATACTCCTTT`; *GAPDH* forward, `GAAGGTGAAGGTCGGAGTC`; and *GAPDH* reverse, `GAAGATGGTGATGGGATTC`.

To examine the latency status of EBV infected cells, expression of *BZLF1* was evaluated with RT-PCR and agarose gel electrophoresis. Primers used were as follows: *BZLF1* forward, `CTGGTGTCCGGGGGATAAT`; *BZLF1* reverse, `TCCGCAGGTGGCTGCT`; *EBER1* forward, `AGGACCTACGCTGCCCTAGA`; *EBER1* reverse, `GGGAAGACAACCACAGACAC`; and *ACTB* forward, `AGAAGGAGATCACTGCCCTGGCACC`; *ACTB* reverse, `CCTGCTTGCTGATCCACATCTGCTG`.

## Western blotting

Whole-cell lysates were extracted from gastric cancer cell lines, and 40 μg of protein was loaded onto a 10% polyacrylamide gel. After electrophoresis and transfer to a polyvinylidene

difluoride membrane, the membranes were incubated with primary and secondary antibodies. The membranes were then incubated with ImmunoStar Reagent (Wako, Osaka, Japan) to visualize protein bands. The primary antibodies used were anti-ARID1A (dilution 1:100; rabbit polyclonal, HPA005456; Sigma-Aldrich) and anti-actin (dilution 1:1000; goat polyclonal antibody; Santa Cruz Biotechnology, Dallas, TX, USA). The secondary antibodies included a peroxidase-conjugated anti-rabbit IgG donkey polyclonal antibody (Jackson ImmunoResearch, West Grove, PA, USA) and a peroxidase-conjugated Fab fragment of an anti-goat IgG rabbit polyclonal antibody (Jackson ImmunoResearch).

### Statistical analyses

Student's t-test was applied to compare continuous variables. Fisher's exact test was used for nominal variables. Statistical analyses were performed with EZR (Saitama Medical Center, Jichi Medical University, Saitama, Japan), which is a graphical user interface for R (The R Foundation for Statistical Computing, Vienna, Australia). More precisely, it is a version of R commander designed to add statistical functions frequently used in biostatistics [19]. *P*-values less than 0.05 was considered statistically significant.

## Results

### EBV-infected epithelial cells and ARID1A expression in the stomach mucosa

To evaluate the relationship between EBV infection and ARID1A-lost expression in non-neoplastic gastric mucosa, screening of EBER-positive cells was performed on archival material. In the first series of screening, there were no EBER-positive glandular cells in the non-tumorous gastric mucosa. However, in the second series, two foci of EBER-positive cells were identified in 1110 blocks from 300 cases (0.18% of tissue blocks and 0.3% of cases). Both lesions harbored a few glands, with EBER-positive epithelial cells showing intestinal metaplasia without morphological atypia (Fig 1A–1C). The background mucosa showed atrophic gastritis with intestinal metaplasia. Four foci in EBER-positive glands were identified during the screening of EBV infection in gastric cancer (Table 1). The carcinoma was EBV-negative in four cases but EBER-positive in two cases. Three lesions, two of which were in the second series, were histologically identified as intestinal metaplastic glands in the lower half of the mucosa propria. The other three were epithelial cells in the upper or middle portion of the mucosa propria showing degenerative or regenerative changes with mild nuclear atypia (Fig 1D–1F).

ARID1A expression was evaluated in serial sections of these six foci of EBER-positive lesions. Two foci could not be evaluated because the EBER-positive foci disappeared during the preparation of serial sections. Three foci showed preserved ARID1A expression, and one showed lost ARID1A. Precise observation of the EBER-positive/ARID1A-lost lesions demonstrated that EBER-positive cells were included in the lesion of ARID1A-lost cells (Fig 1E and 1F). Lost ARID1A was not observed in EBER-negative glands in these four sections analyzed by immunohistochemistry.

### ARID1A-lost and -preserved EBVaGCs in intramucosal cancer

ARID1A immunohistochemistry was performed on 12 intramucosal EBVaGC from 11 cases (Table 2), demonstrating seven ARID1A-preserved and five ARID1A-lost tumors (Fig 2). One case (Case 7) had two intramucosal EBVaGCs. ARID1A-lost EBVaGC localized in the upper portion of the stomach in four of five cases, while ARID1A-preserved EBVaGC was present in two of seven cases. The tumors appeared to differ in their histological features. ARID1A-lost

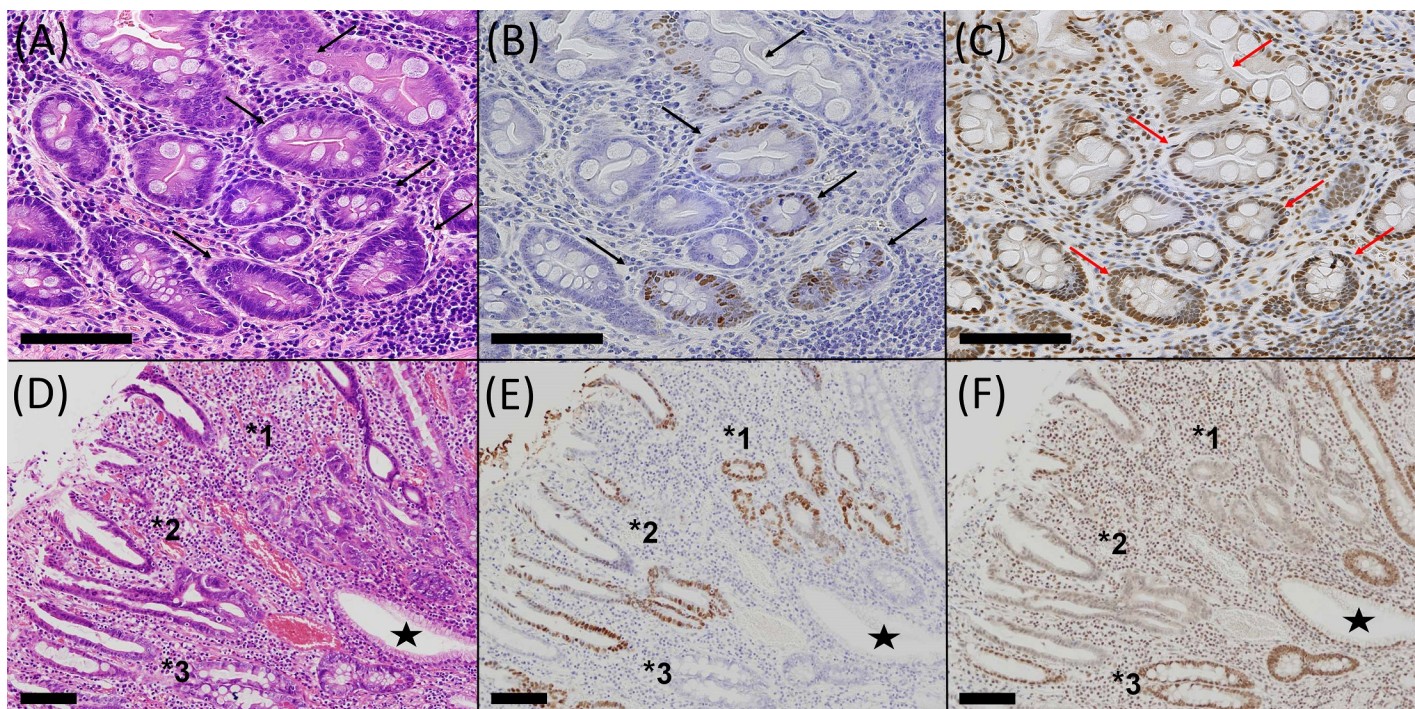

**Fig 1. EBER-positive epithelial cells in non-neoplastic mucosa.** Two types of EBER-positive cells are shown: the metaplastic glandular type (A–C) and the surface degenerative type (D–F). Glandular cells of intestinal metaplasia without morphological atypia (arrows, A) show positive nuclear signals for EBER (arrows, B). Of note, these cells retained nuclear ARID1A expression (arrows, C). Surface epithelial cells in the upper or middle portion of the mucosa propria with degenerative changes and mild nuclear atypia (*1–3, D) show positive nuclear signals for EBER (*1–3, E). The nuclei of these cells were negative for ARID1A (*1, 3, F), and EBER-positive cells were included in the lesions of ARID1A-lost epithelial cells (*2, E–F). Asterisks indicate control epithelial cells showing EBER-negative and ARID1A-positive nuclei. A and D, hematoxylin–eosin staining; B and E, EBER *in situ* hybridization; and C and F, ARID1A immunostaining. Scale bar, 100μm.

EBVaGC showed a tubular formation more distinctly, and ARID1A-preserved EBVaGC predominantly showed a cord-like structure. When the grade of luminal formation was scored as 0–3 (0, none; 1, <10%; 2, 10%–50%; and 3, >50% of cancer cell nests), four of five ARID1A-lost EBVaGCs showed grade 2–3, while five of seven ARID1A-preserved tumors showed grade 0–1 (p = 0.0659, Wilcoxon rank sum test). EBER-positive cells corresponded to ARID1A-lost cells in almost all cases in ARID1A-lost EBVaGC. However, lost ARID1A was observed in EBER-negative non-neoplastic mucosa, which was just adjacent to the cancer tissue of EBVaGC in Case 9 (Fig 2E and 2F).

**Table 1. Characteristics of EBER-positive non-tumorous glands in the background gastric mucosa.**

| Case | Age (years) | Sex | Main tumor | Mucosa | Histology of EBER-positive glands | ARID1A in EBER-positive foci |
|---|---|---|---|---|---|---|
| 1 | 67 | M | EBV-negative | PG + FG + IM | Intestinal metaplastic glands | Retained |
| 2 | 59 | M | EBV-negative | FG + IM | Intestinal metaplastic glands | Retained |
| 3 | 55 | M | EBV-negative | PG + IM | Degenerative epithelia | Lost |
| 4 | 53 | F | EBV-negative | IM | Degenerative epithelia | N.A. |
| 5 | 73 | M | EBVaGC | PG + FG + IM | Degenerative epithelia | N.A. |
| 6 | 37 | M | EBVaGC | PG + IM | Intestinal metaplastic glands | Retained |

Cases 1 and 2 were detected in the screening of 2246 blocks. Cases 3 to 6 were incidentally detected in the previous studies.

PG, pyloric or pseudopyloric glands; FG, fundic glands; IM, intestinal metaplasia.

N.A., not assessed (in Cases 4 and 5, the lesion had disappeared after the re-cutting of sections for immunohistochemistry).

**Table 2. Clinicopathological characteristics of intramucosal EBVaGC with preserved or lost ARID1A.**

| Case Number | ARID1A expression | Sex | Age (years) | Location | Gross type | Tumor size (mm) | Histology and luminal formation score | | Multiplicity* the other carcinoma |
|---|---|---|---|---|---|---|---|---|---|
| 1 | Preserved | M | 60 | M | IIb | 15 | Cord-like/signet ring-cell | 0 | |
| 2 | Preserved | M | 37 | L | IIc | 8 | Cord-like | 0 | |
| 3 | Preserved | M | 59 | M | IIc | 33 | Tubular | 3 | EBVaGC (SM, ARID1A-lost) |
| 4 | Preserved | M | 76 | M | IIc+a | 25 | Cord-like | 0 | |
| 5 | Preserved | M | 77 | M | IIc+a | 45 | Cord-like/lace | 1 | |
| 6 | Preserved | F | 55 | U | IIc | 30 | Tubular/cord-like/lace | 2 | |
| 7–1 | Preserved | F | 46 | U | IIb | 30 | Cord-like/tubular | 1 | EBVaGC (Case 7–2) |
| 7–2 | Lost | | | U | IIc | 30 | Tubular/cord/lace | 2 | EBVaGC (Case 7–1) |
| 8 | Lost | M | 64 | M | IIc | 25 | Cord-like/tubular/lace | 1 | |
| 9 | Lost | M | 69 | U | Is | 18 | Tubular/cord | 3 | |
| 10 | Lost | F | 73 | U | Is | 12 | Tubular | 3 | EBVaGC (SM, ARID1A-lost) |
| 11 | Lost | M | 60 | U | IIc | 24 | Tubular/lace/cord (3) | 3 | |

Location and gross subtype are presented according to the Japanese Classification of Gastric Cancer (2nd ed.). U, upper 1/3 of the stomach; M, middle; L, lower.

Luminal formation is scored as follows: 0, none; 1, <10%; 2, 10%–50%; 3, >50%. SM, submucosal invasion.

*When there are two gastric carcinomas in the same case, the characteristics of the other carcinoma was described in the "Multiplicity" column.

## EBV infection efficiency in *ARID1A*-knockdown gastric cancer cell lines

To evaluate the effect of lost ARID1A on the efficiency of EBV infection, *ARID1A* was knocked down with siRNA in the EBV-negative gastric cancer cell lines MKN74 and NUGC3, followed by cell-to-cell contact infection using the Akata infection system. *ARID1A* knockdown significantly increased the number of EBV-infected colonies by 3-fold in MKN74 cells and 6-fold in NUGC3 cells after incubation with G418 (Fig 3).

## EBV infection efficiency in *ARID1A*-knockout gastric cancer cell lines

*ARID1A* was knocked out in MKN74 cells using the CRISPR Cas9 system, and two clones with lost ARID1A were obtained (Fig 4A, right panel). *ARID1A* knockout induced a fibroblastic morphology with filopodia formation of various degrees (Fig 4A). These cells were infected with cell-free EGFP-EBV. In both EBV-infected MKN74 parental cells and *ARID1A*-knockout MKN74 clones, expression of *BZLF1* was not observed, confirming the latent infection status (Fig 4B and S1 Fig). Both *ARID1A*-knockout MKN74 clones #1 and #2 yielded substantially more EGFP-positive cells compared with the parental MKN74 cell line (Fig 4C). The increase in EGFP-positive cells was further confirmed by flow cytometry analysis, which demonstrated 15- and 14-fold increases in clones #1 and #2, respectively (Fig 4D).

## Discussion

Complex interaction between the virus and host is the primary issue in virus-associated neoplasms from the initial to the advanced stages of cancer development, which we termed the "gastritis infection carcinoma sequence of EBVaGC" [13]. In the present study, we confirmed the co-existence of EBV infection and lost ARID1A in non-neoplastic epithelial cells of the stomach. In the second series of screening, the frequency of EBER-positive glands was 0.2% in non-neoplastic mucosa tissues. As a reference for frequencies of precursor lesions in non-neoplastic mucosa, we could take an example of *TP53* gene abnormalities in gastric carcinogenesis, although mutation of *TP53* is an early event in gastric carcinogenesis of the chromosomal

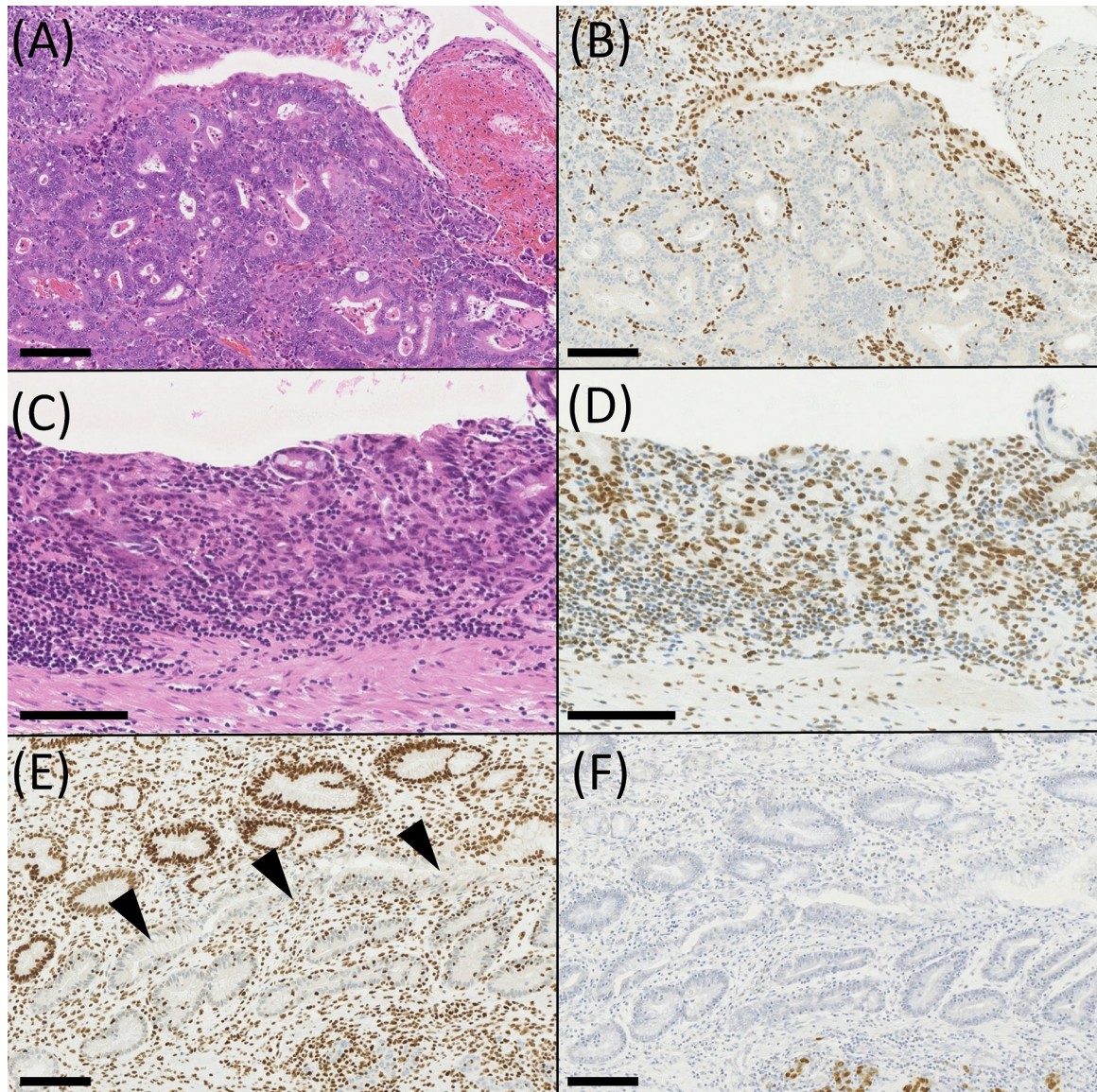

**Fig 2. Histology of intramucosal carcinoma of EBVaGC.** (A, B) Histology of EBVaGC with lost ARID1A, showing well-formed distinct glands with preserved lumen. (C, D) Histology of EBVaGC with preserved ARID1A. The lumen of glands was indistinct, and abortive glands and a cord-like pattern was observed. (E, F) A case of EBVaGC with lost ARID1A. EBER-negative non-neoplastic mucosa adjacent to the tumor (arrowhead) showed lost ARID1A. A and C, hematoxylin–eosin staining; B, D, and E, immunohistochemistry of ARID1A; F, EBER *in situ* hybridization. Scale bar, 100μm.

instability subtype of gastric cancer (nearly 50% of total gastric cancer) [2, 20]. During the search for p53-positive cells in non-neoplastic mucosa tissues of the stomach, Ochiai et al. identified 19 foci in 756 tissue sections (2.5%) [21]. Since EBVaGC comprises 3.5–10% of gastric cancer with a half showing loss of ARID1A, the relative incidence of p53-positivity and EBV infection in non-neoplastic mucosa (25:2) appeared to correspond with that in gastric cancer, suggesting that both are precursors of each subtype of gastric cancer. It is worth noting that EBER-positive epithelial cells were relatively frequently identified in the proximity of gastric cancer with or without EBV infection, suggesting that field cancerization plays a role in the carcinogenesis of EBVaGC.

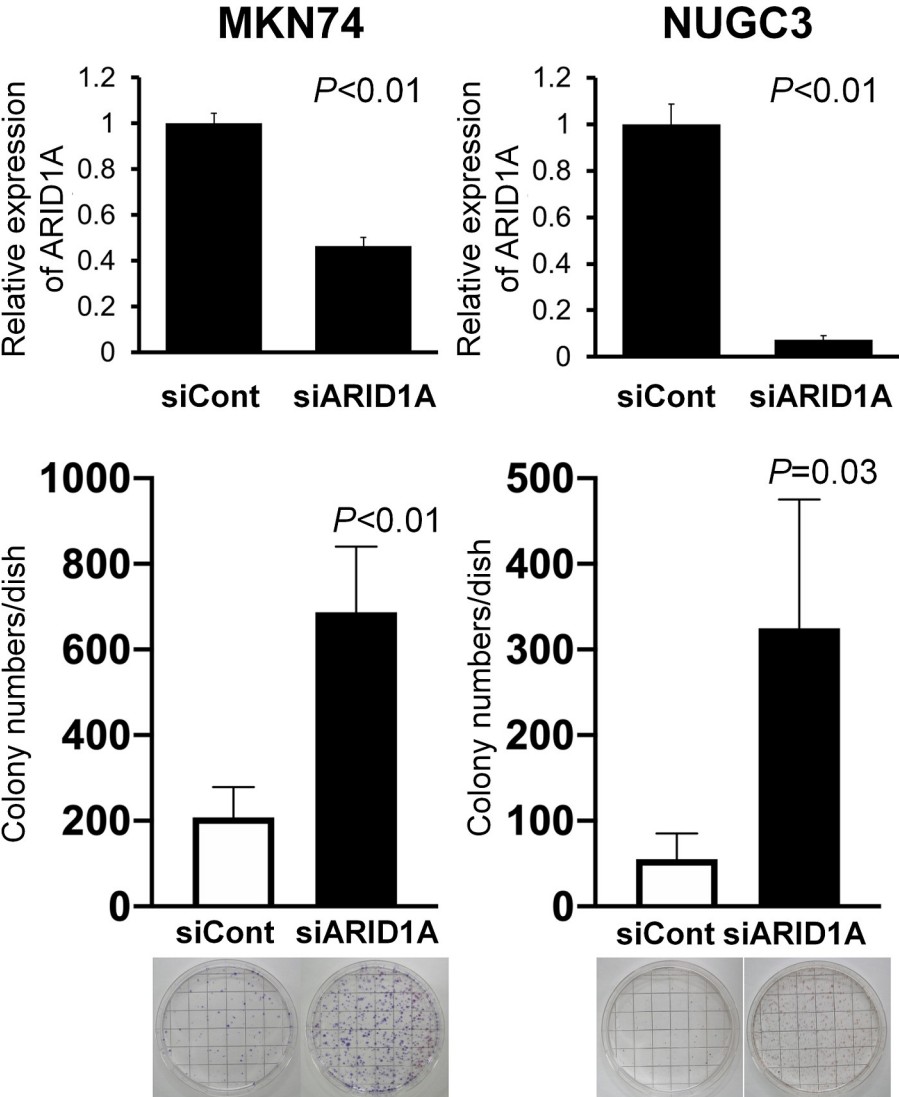

**Fig 3. EBV infection of *ARID1A*-knockdown gastric cancer cells.** MKN74 and NUGC3 gastric cancer cells were incubated with siRNA targeting *ARID1A* mRNA and subjected to cell-to-cell contact infection with EBV harboring the neomycin-resistant gene. After selection with G418, resistant colonies were stained with hematoxylin. Note a three- and six-fold increase in G418-resistant colonies in *ARID1A*-knockdown MKN74 and NUGC3 cells, respectively, compared with the cells treated with control siRNA.

In our previous study, lost ARID1A was correlated with no clinicopathological factors or with adverse patient prognosis [11]. However, in the present study of the intramucosal stage of EBVaGC, there were morphological differences between ARID1A-lost and -preserved carcinomas. There was a tendency that the former showed distinct glands, while the latter showed a cord-like structure. These facts suggest that EBER-positive glands with lost and preserved ARID1A expression in the non-neoplastic mucosa are direct precursor of ARID1A-lost and ARID1A-preserved carcinomas of EBVaGC, respectively.

A comparative analysis of EBER-ISH and ARID1A immunohistochemistry demonstrated that lost ARID1A was observed in the non-neoplastic mucosa adjacent to the tumor. In EBER-positive lesions of non-neoplastic mucosa, we could examine the distribution of EBER-positive

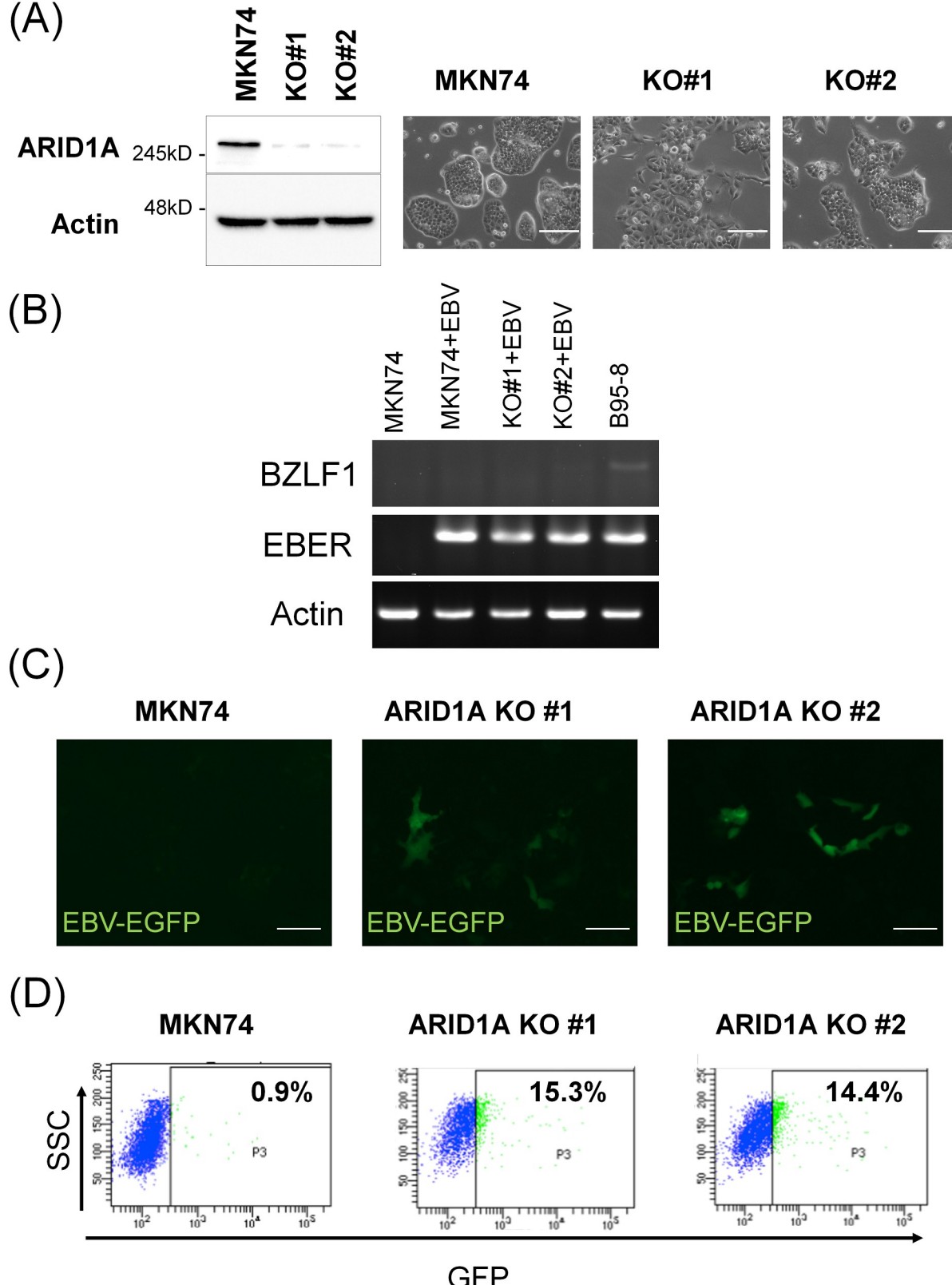

**Fig 4. EBV infection of *ARID1A*-knockout gastric cancer cells.** (A) The *ARID1A* gene was knocked out in the gastric cancer cell line MKN74 using CRISPR Cas9. Two *ARID1A*-knockout clones show a fibroblastic morphology of various degrees. (B) RT-PCR of BZLF1 gene

of EBV, showing absence of expression in both EBV-infected parental MKN74 cells and ARID1A-knockout clones. (C) Parental MKN74 cells and *ARID1A*-knockout MKN74 clones #1 and #2 were infected with GFP-integrated cell-free EBV. Several *ARID1A*-knockout cells showed green fluorescence on Day 2. (D) Flow cytometry analyses of fluorescent-positive cells on Day 2. SSC, side scatter; FITC, fluorescein isothiocyanate.

cells and ARID1A-lost cells on a gland-by-gland basis, and EBER-positive cells were included in ARID1A-lost cells. Both facts further suggest that lost ARID1A precedes EBV infection.

To test the possible contribution of lost ARID1A to the EBV infection of stomach epithelial cells, we infected *ARID1A*-knockdown and -knockout gastric cancer cells with EBV. In both *ARID1A*-knockdown and *ARID1A*-knockout cells, the infection efficiency was markedly increased by 3- to 6-fold and 15-fold, respectively. The difference in the extent of this increase may be because of the different methods used to reduce ARID1A expression or infect cells with EBV. We used the classical cell-to-cell contact method in the former experiment but the EGFP-EBV method in the latter. It took longer to obtain the results in the former. Alternatively, with G418 selection process, the establishment of infection might be unsuccessful in a certain proportion of EGFP-positive cells in the latter. In fact, EBV could escape from infected epithelial cells during cell culture [22]. Along the line of thinking, loss of ARID1A might contribute to keep EBV in infected cells through unknown mechanisms. Tsai et al. recently demonstrated that ARID1A loss causes decrease of recruitment of genome stability regulators such as ATR and TOP2 [23]. Further studies are also necessary to clarify interaction of EBV and chromosomal modulators [15]. Nevertheless, the results of EBV infection of *ARID1A*-knockdown and *ARID1A*-knockout cells are consistent with the comparative observation of EBER *in situ* hybridization and ARID1A immunohistochemistry.

In the present study, we demonstrated the following: 1) lost ARID1A coexisted with EBV-infected cells in non-neoplastic epithelial cells showing degenerative glands; 2) lost ARID1A correlated with morphological features (tubular structure) of EBVaGC in the proper mucosal layer; and 3) lost ARID1A promoted the infection of EBV in stomach epithelial cells, indicating that lost ARID1A induces permissive conditions for EBV infection, thereby initiating the viral-driven carcinogenesis. Lost ARID1A is one of the constituents of virus–host interactions in the carcinogenesis of EBVaGC in addition to "cross-species DNA hypermethylation" [14] and "cross-species chromosomal interactions" [15].

## Supporting information

**S1 Fig. Gel images for Fig 4B.**
(TIF)

## Acknowledgments

The authors thank Keisuke Matsusaka (Department of Pathology, Chiba University) for his excellent advice, and Kei Sakuma, Minato Murata, Kimiko Takeshita, and Tomoko Irisa for their excellent technical assistance. We also thank Melissa Crawford and H. Nikki March, PhD, from Edanz Group (https://en-author-services.edanzgroup.com/ac) for editing a draft of this manuscript.

## Author Contributions

**Conceptualization:** Hiroyuki Abe, Atsushi Kaneda, Masashi Fukayama.

**Data curation:** Hiroyuki Abe, Akiko Kunita, Yuya Otake.

**Funding acquisition:** Hiroyuki Abe, Atsushi Kaneda, Masashi Fukayama.

**Investigation:** Yuya Otake.

**Methodology:** Teru Kanda.

**Supervision:** Tetsuo Ushiku, Masashi Fukayama.

**Writing – original draft:** Hiroyuki Abe.

**Writing – review & editing:** Masashi Fukayama.

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
