## [Decision Letter · Decision Letter 0]

8 Mar 2021

PONE-D-21-04978

Virus–host interactions in carcinogenesis of Epstein-Barr virus-associated gastric carcinoma: Potential roles of lost ARID1A expression in its early stage

PLOS ONE

Dear Dr. Fukayama,

Your manuscripts are reviewed by two experts in EBV fields.  I also read your manuscript quickly. We all agree that this is an interesting and potentially important work.  However, they also identified several significant concerns that require further attention. As indicated in the appended comments, a number of specific points have been raised, especially Reviewer #2, that have to be resolved. Other than the Reviewers’ comments, I would like to ask you to address the EBV infection status, i.e, lytic and latency, in those ARID1A modulated specimens and EBV-infected cells in vitro.  Also, why you used two different methods for EBV infection?  Based on the combined assessments of the reviewers, we would be willing to provide you with an opportunity to respond to these issues in a suitably revised version of the manuscript.  After careful consideration, we feel that your manuscript has merit and will be reconsidered for publication in PLoS One after major revision.

We look forward to receiving your revised manuscript.

Kind regards,

Luwen Zhang

Academic Editor

PLOS ONE

Journal Requirements:

4. Please include your tables as part of your main manuscript and remove the individual files. Please note that supplementary tables should be uploaded as separate "supporting information" files.

Reviewers' comments:

Reviewer's Responses to Questions

**Comments to the Author**

1. Is the manuscript technically sound, and do the data support the conclusions?

Reviewer #1: Yes

Reviewer #2: Yes

2. Has the statistical analysis been performed appropriately and rigorously? 

Reviewer #1: Yes

Reviewer #2: Yes

3. Have the authors made all data underlying the findings in their manuscript fully available?

Reviewer #1: Yes

Reviewer #2: Yes

4. Is the manuscript presented in an intelligible fashion and written in standard English?

Reviewer #1: Yes

Reviewer #2: Yes

5. Review Comments to the Author

Reviewer #1: The authors showed an important role of ARID1A for EBV infection into gastric epithelial cells. They clearly elucidated that lost ARID1A increased the efficiency of EBV infection to gastric cells in vitro. Their results may explain that ARID1A is frequently lost in EBV-associated gastric cancer.

I have several questions.

1. Is G418-resistant colony composed of 100% EBNA-positive cells? Could you confirm EBV infection in each cell like Figure 4B?

2. As shown in Figure 1 E and F, I agree that the expression of EBER and ARID1A is not coincident. The incidence is quite low. Is it a significant finding in EBVaGC?

3. Are their any morphologic changes in ARID1A-KO cells compared to the parent cells?

4. “BVaGC” means EBVaGC in Table 2?

5. Have you examined alterations in the ARID1A gene?

Reviewer #2: The authors state that loss of AIDA1A enhance EBV infection in gastric mucosa, and eventually, development of EBVaGC. The key concept is interesting and their strenuous data seems suggestive for the statement. However, interpretation of clinicopathological examination data is not easy for me.

Specific comments

Table2

#1:

Q: Case3; ARID1A-preserved is mentioned to EBVaGD (SM, ARID1A-lost. What does this mean?

Q: Case7; Are BVaGC typo?

Abstract

#2: “Screening of EBER by in situ hybridization revealed a frequency of approximately 0.2% EBER-positive epithelial cells in non-neoplastic gastric mucosa tissue samples. Six small foci of EBV-infected epithelial cells showed two types of histology: degenerated (n=3) and metaplastic (n=3) epithelial cells. One of the degenerated EBER-positive foci was located within the ARID1A-lost glands. ARID1A-lost EBVaGC (n=5) showed a tubular structure with distinct luminal formation compared with ARID1A-preserved cancer (n=7) in the proper mucosal layer, suggesting morphological continuity of both EBER-positive/ARID1A-lost epithelial cells.”

Q: It is hard to understand the significance of this statement for general readers. I can’t follow the relationship of morphological continuity with EBER/ARID1A status.

Materials and Methods

#3: Tissue Samples:

Q: How did the authors select three series of samples for EBERs-screening? Also, they mention the detail of first and second series. However, they did not describe the detail for the third series. Why?

Results

#4: ARID1A expression was evaluated in serial sections of these six foci of EBER-positive lesions. Two foci could not be evaluated because the EBER-positive foci disappeared during the preparation of serial sections. Three foci showed preserved ARID1A expression, and one showed lost ARID1A.

Q: According to the authors’ statement, ARID1A expression precede before EBV-infection. However, 3 of 4 EBER-positive cases retained ARIDA1A expression. Please explain.

#5: “ARID1A-lost EBVaGC showed a tubular formation more distinctly, and ARID1A-preserved EBVaGC predominantly showed a cord-like structure. Lace pattern, which is an interconnecting cord-like or tubular structure [19], was observed in both ARID1A-lost and -preserved EBVaGC”

Q: It is hard to understand the relevance of tissue structure and ARID1A, EBERs in EBVaGC.

Please explain plainly.

Discussion

#6:

Q: Relationship of TP53, ARID1A, and EBVaGC is not clear for me. Also, morphological relevance of ARID1A and EBER-positive cell in terms with EBV-associated carcinogenesis is not clearly described. According to the authors’ statement, ARID1A mutation, which precede to EBV infection, should be more frequent than TP53 mutation. Their statement is quite confusing to me. Please simplify the story.

#7:

Q: They show ARID1A-lost gastric cancer cell lines gained infection efficiency. I think the data itself is interesting. However, EBV genome is frequently lost, especially in the EBV-infected epithelial cells. So it is suggested that some factors that keep EBV in the infected cell is important for the persistent infection. So, EBV might have escaped from certain proportion of gastric epithelial cells. Do you have any comments?

6. PLOS authors have the option to publish the peer review history of their article (what does this mean?). If published, this will include your full peer review and any attached files.

Reviewer #1: No

Reviewer #2: No

---

## [Author Response · Author response to Decision Letter 0]

22 Jun 2021

Response to each comment of the reviewers and the editor is described in the "Response to Reviewers" file.

---

## [Decision Letter · Decision Letter 1]

9 Aug 2021

Virus–host interactions in carcinogenesis of Epstein-Barr virus-associated gastric carcinoma: Potential roles of lost ARID1A expression in its early stage

PONE-D-21-04978R1

Dear Dr. Fukayama,

We’re pleased to inform you that your manuscript has been judged scientifically suitable for publication and will be formally accepted for publication once it meets all outstanding technical requirements.

Kind regards,

Luwen Zhang

Academic Editor

PLOS ONE

Additional Editor Comments (optional):

Reviewers' comments:

Reviewer's Responses to Questions

**Comments to the Author**

1. If the authors have adequately addressed your comments raised in a previous round of review and you feel that this manuscript is now acceptable for publication, you may indicate that here to bypass the “Comments to the Author” section, enter your conflict of interest statement in the “Confidential to Editor” section, and submit your "Accept" recommendation.

Reviewer #2: All comments have been addressed

2. Is the manuscript technically sound, and do the data support the conclusions?

Reviewer #2: Yes

3. Has the statistical analysis been performed appropriately and rigorously? 

Reviewer #2: Yes

4. Have the authors made all data underlying the findings in their manuscript fully available?

Reviewer #2: Yes

5. Is the manuscript presented in an intelligible fashion and written in standard English?

Reviewer #2: Yes

6. Review Comments to the Author

Reviewer #2: I think they have addressed to the comments properly, and now acceptable for publishing on PLOS One..

7. PLOS authors have the option to publish the peer review history of their article (what does this mean?). If published, this will include your full peer review and any attached files.

Reviewer #2: No